# Human lipoproteins comprise at least 12 different classes that are lognormally distributed

Tomokazu Konishi[1]*, Risako Fujiwara[2,3], Tadaaki Saito[1], Nozomi Satou[1], Yurie Hayashi[1], Naoko Crofts[1], Ikuko Iwasaki[1], Yoshihisa Abe[3], Shinpei Kawata[2], Tatsuya Ishikawa[2]

1 Graduate School of Bioresource Sciences, Akita Prefectural University, Akita, Japan, 2 Research Institute of Akita Cerebrospinal and Cardiovascular Center, Akita Prefectural Hospital Organization, Akita, Japan, 3 Cardiovascular internal medicine, Akita City Hospital, Akita, Japan

* konishi@akita-pu.ac.jp

**Data Availability Statement:** Data relevant to this study are available in figshare from https://doi.org/10.6084/m9.figshare.14247119.v1.

## Abstract

This study presents the results of HPLC, a gentler and rapid separation method in comparison with the conventional ultracentrifugation, for 55 human serum samples. The elution patterns were analysed parametrically, and the attribute of each class was confirmed biochemically. Human samples contained 12 classes of lipoproteins, each of which may consist primarily of proteins. There are three classes of VLDLs. The level of each class was distributed lognormally, and the standard amount and the 95% range were estimated. Some lipoprotein classes with a narrow range could become ideal indicators of specific diseases. This lognormal character suggests that the levels are controlled by the synergy of multiple factors; multiple undesirable lifestyle habits may drastically increase the levels of specific lipoprotein classes. Lipoproteins in medical samples have been measured by enzymatic methods that coincide with conventional ultracentrifugation; however, the high gravity and time required for ultracentrifugation can cause sample degradation. Actually, the enzymatic methods measured the levels of several mixed classes. The targets of enzymatic methods have to be revised.

## Introduction

Lipoproteins are measured in two ways. The first is related to class separation for biochemical purposes [1–5]. This method uses ultracentrifugation, which spontaneously creates a salt density gradient due to centrifugal forces. Although the classes of lipoproteins are separated according to their density, this method is time-consuming and is not suitable for measuring large numbers of medical samples. The other method, which uses enzymatic processes, is used for physical examination [6]. In this method, cholesterol is chemically extracted from a certain class of lipoproteins and then measured. Several kits are available, but all are adjusted to mimic the results of the ultracentrifugation method. In the ultracentrifugation approach, the larger the particle size, the lower the density of lipoproteins; therefore, names such as very low-

**Funding:** TK was funded by Akita Prefectural University.

**Competing interests:** The authors have declared that no competing interests exist.

density lipoprotein (VLDL) and high-density lipoprotein (HDL) are used for large and small particles, respectively [7,8].

The accuracy of ultracentrifugation has been questioned in a study using rat serum [9]. The very high centrifugal force was sufficient to pull hydrophobic proteins out of the membrane [10]; additionally, complete separation can take several days, during which proteins can be degraded. There is a gentler way to separate these classes via HPLC gel filtration. This takes up to 30 min and does not require extra salts [11]. HPLC is not a novel method, but there were a few issues with how the data were analysed, as it was noted that the data were intended to be consistent with standard ultracentrifugation results. Analysis of HPLC results of rat data with parametric analysis (Materials and Methods) showed striking differences from the ultracentrifugation results [9]. All classes of lipoproteins are protein-rich particles, contrary to conventional knowledge [5,7,8]. Two new classes, LDL-antiprotease complexes (LAC), were also discovered.

## Materials and methods

Blood samples: Samples were collected after obtaining informed consent from all volunteers and approval from the ethics committee of Akita Cerebrospinal and Cardiovascular Center (ID. 19–21). All samples were anonymized prior to analysis. No postmeal time was specified for blood collection. The age of the volunteers were shown in S1 Fig in S1 File. The ratio of men to women was 1:1.

The collected blood was sent to a clinical laboratory (SRL Inc., Tokyo, Japan). The serum was separated, and sent to Skylight Biotech Inc. (Akita, Japan) for further analysis using gel filtration HPLC [11]. HDL, LDL, and total cholesterol were measured using conventional enzymatic methods used in the clinical laboratory [12,13]. Forty-four healthy volunteer samples were subjected to analytical HPLC, and TG and cholesterol were monitored sequentially. Furthermore, 6 samples were subjected to preparative HPLC and then fractionated. There were no chylomicronaemic samples.

The HPLC monitoring data (TG and cholesterol) were analysed parametrically [9]. This method is a parsimonious way of performing curve fit to maintain the falsifiability of the model using the minimum number of classes assumed (S2 Fig in S1 File) [14]. With many estimated classes, the fitting process will become easier; however, the assumed classes must be verified by reality. Too many assumptions make this verification difficult. The size and range of a class are presented using the position μ and scale σ of the normal distribution. We assumed that a class would contain TG and cholesterol at a certain constant rate regardless of the size differences within the class., with The amount of each TG and cholesterol presented by using another parameter. This assumption was verified through the curve-fitting process.

The standard values of TG and cholesterol in each class were estimated from the full data set using a trimmed mean (0.2). Their 95% range was estimated using the median absolute deviation (MAD): the upper and lower limits of the 44 healthy samples were estimated as trimmed mean to two MADs. The standard values of the position or scale parameters of the classes, which were varied, were estimated from the trimmed mean of μ or $\sigma^2$ found in each sample.

In preparative HPLC, the elution was periodically fractionated. Each fraction was subjected to 5%–20% SDS-PAGE, and the proteins were detected using silver staining. Some protein bands were identified using MALDI-TOF MS (Genomine Inc., Kyungbuk, Korea) [15]. In addition, specific proteins were confirmed by western blotting after transfer to PVDF membranes. The antibodies used were as follows: anti-apoB antibody (A-6), sc-393636 AF488; apoA-I antibody (B-10), sc-376818 AF647 (Santa Cruz Biotechnology Inc., Texas, USA); anti-

Lipoprotein a antibody, ab27631, (Abcam plc., London, UK). Chemiluminescence of the antibodies and silver-stained gel bands were measured using an Amersham Typhoon Scanner (Cytiva). The concentrations of proteins in the HPLC fractions, which had been diluted inevitably, were determined using the BCA assay (Takara, Shiga Japan), which bases on the same reaction but has better sensitivity than the biuret reaction that is commonly used for serum samples.

## Results

At least 12 classes of lipoproteins in normal distribution had to be presumed to fit the human data (Fig 1 and S3 Fig in S1 File). At least 12 classes of lipoproteins in the normal distribution were presumed to fit the human data (Fig 1 and S3 Fig in S1 File). The attributes of the classes were determined based on the elution pattern of the major protein components (Figs 1 and 2, and S4 Fig in S1 File); even a minor class cannot be ignored (S2 Fig in S1 File). In Fig 1 and S4 Fig in S1 File, the position of the bands in the superimposed SDS-PAGE photograph roughly corresponds to the elution time in the background graph. Moreover, the lanes of the SDS-PAGE correspond to the fraction numbers marked directly below. The elution patterns of these proteins coincided exactly with the distribution of the corresponding classes (Fig 2 and S4 Fig in S1 File). Additionally, TG and cholesterol had coincident positions and coincident scales in each class. Those are phenomena that are only observed if the estimated class of particles are actually present and if they maintain a constant content, regardless of size fluctuation. Those also suggest that the curve fitting was performed properly. When two classes with different TG/cholesterol ratios have similar diameters, and if they form a single composite

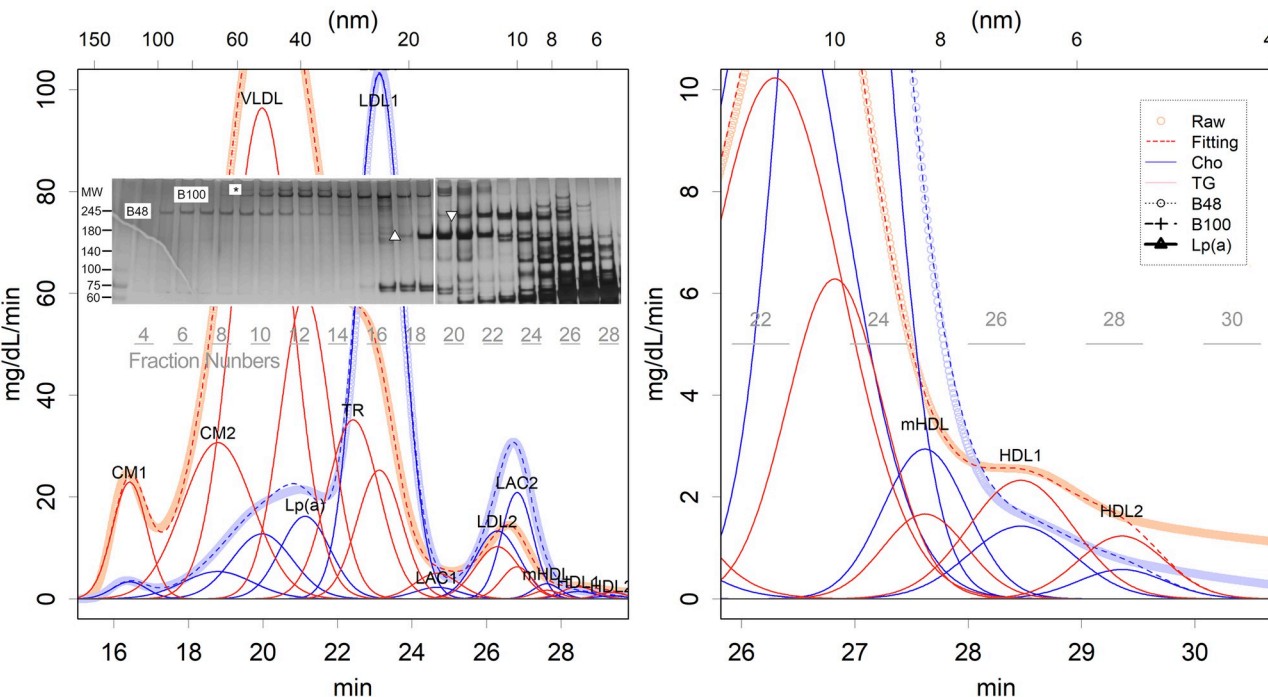

**Fig 1. Elution pattern of HPLC of a hyperlipidaemia patient.** In gel filtration, the shorter the elution time, the larger the particle size; elution time is proportional to the logarithms of the diameter of the particles. Left: Whole image, right: Enlarged view around HDLs. The bold line presents the measurement raw data, the red (TG) and blue (cholesterol) lines are the curve-fitted classes, and the dotted lines are their sum. There are 12 classes. The superimposed photo is a part of SDS-PAGE. Each lane corresponds to the fraction number displayed directly below it (gray). B100 and B48 are the respective ApoB positions. △:Alpha-2-macroglobulin, ▽: Inter-alpha-trypsin inhibitor heavy chain, *: Lp(a). Here an example of a hyperlipidaemic patient is shown so that the classes can be easily observed. An alternative to healthy volunteers is shown in S3 Fig in S1 File.

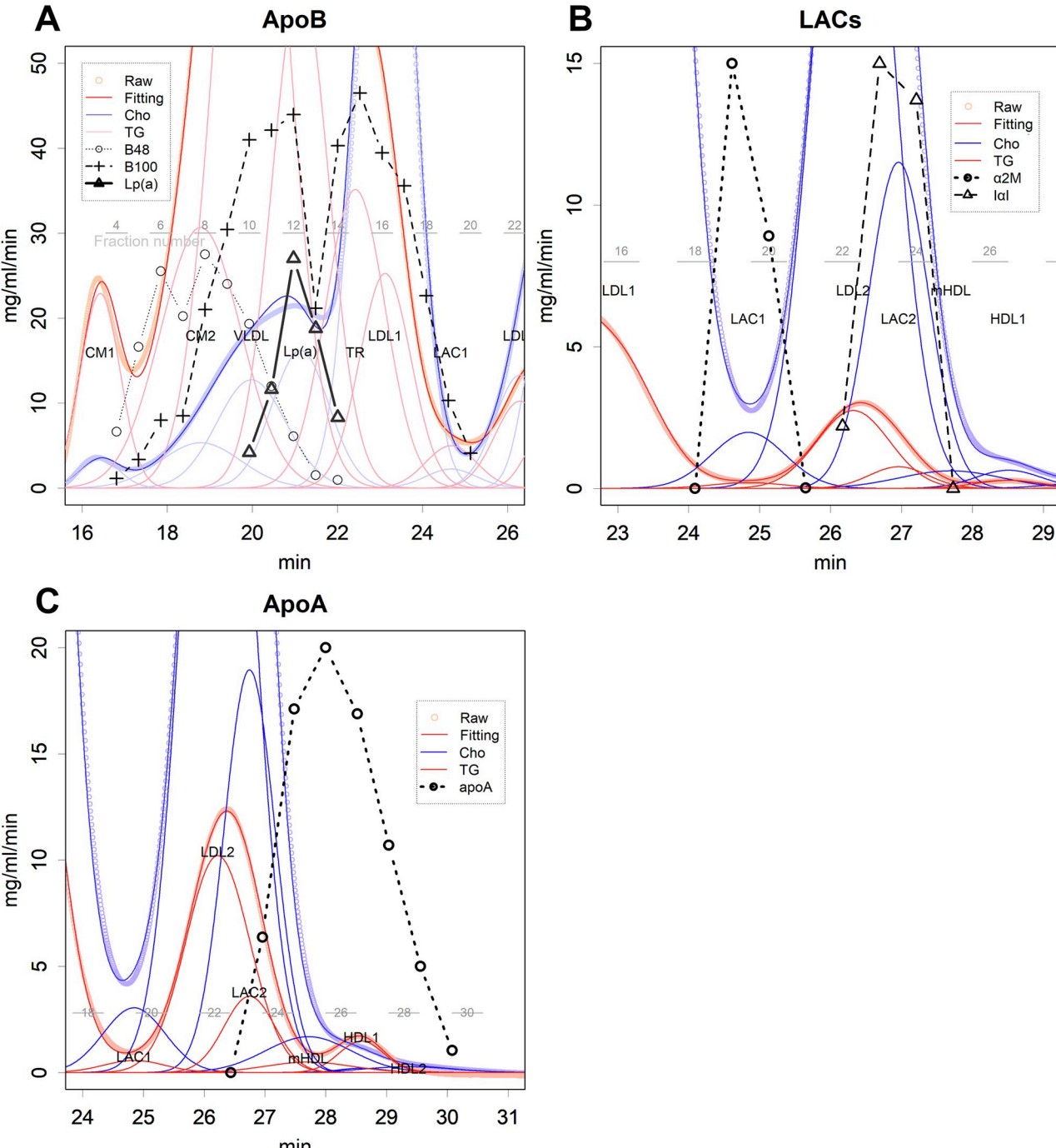

**Fig 2. Elution pattern of proteins. A**. ApoB100, ApoB48, and Lp(a). **B**. LACs. Bands of the silver-stained SAS-PAGEs were determined by densitometry. The bands were identified by MALDI-TOF MS and western blotting using specific antibodies. The grey horizontal numbers and lines indicate the fraction numbers collected in the HPLC separation. **C**. Western blot results for ApoA.

peak, then the TG and cholesterol peak times will be different. This is actually seen in the relationship between LDL2 and LAC2; the CH and TG of these particles form a composite peak, but the Ch peak appeared later (Fig 1). The LAL2 has more cholesterol (Fig 3A) and is slightly smaller, causing a delay in the appearance of the cholesterol peak.

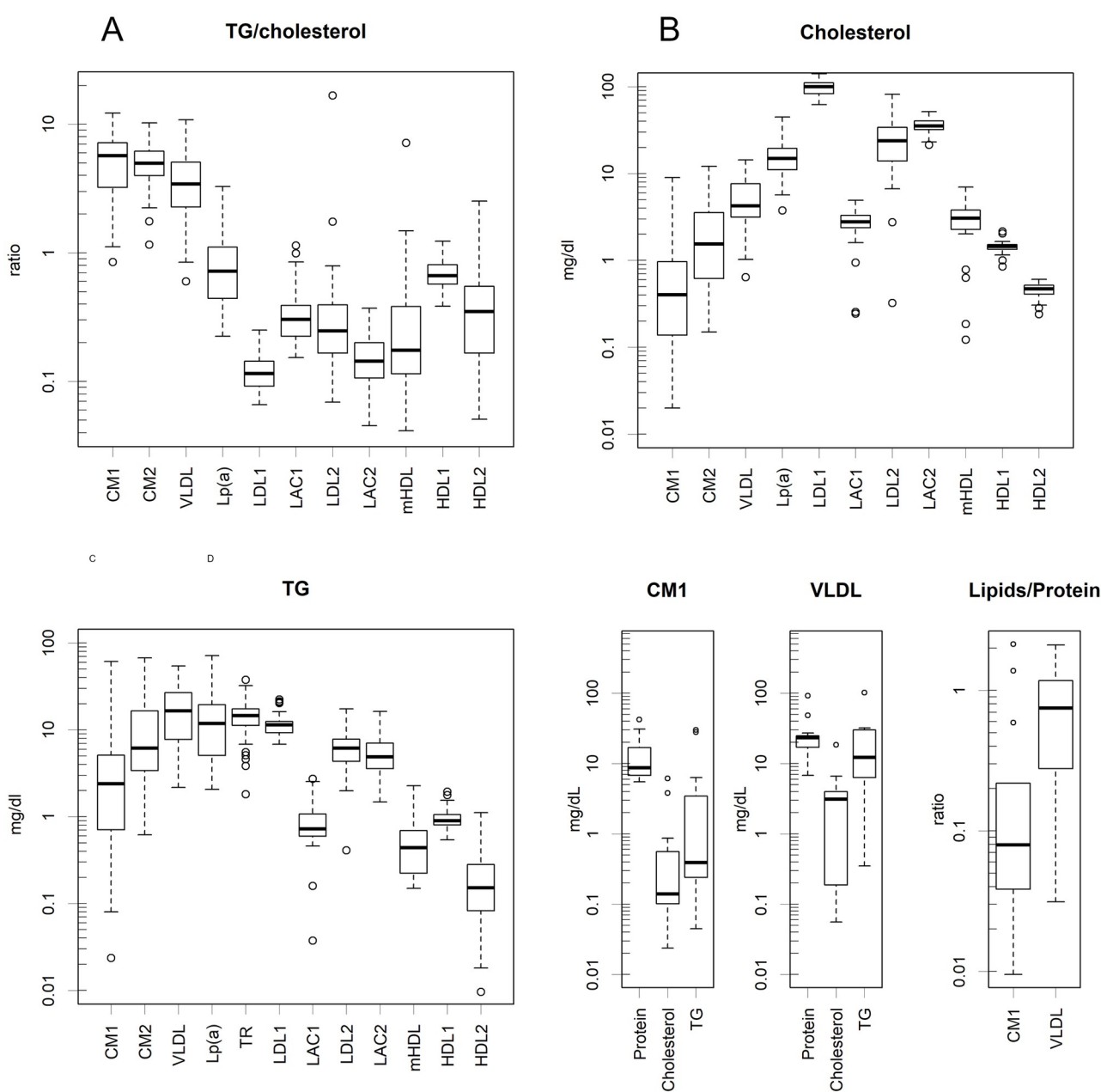

**Fig 3. Box plot of the logarithm of quantity.** All the y-axes are on a logarithmic scale. **A**. The ratio of TG/cholesterol. Naturally, the ones produced with TG are higher. The contents are **B**. Cholesterol and **C**. TG (mg / dL). While LDL1, LACs, and HDLs appeared within a certain narrow range, CM and VLDL levels fluctuated. **D**. Amounts and ratios of CM and VLDL measured by the preparative HPLC of healthy samples. Lipids (TG + cholesterol) are less than proteins in those classes.

CM and VLDL were eluted, consistent with observed peaks of ApoB48 and ApoB100, respectively (Figs 1A and 2A and S4A in S1 File). Most ApoB proteins appear to be degraded during TG removal. In fact, the LDL1 fraction contains less B100 than expected for a large number of particles, and the LDL2 fraction is devoid of B48. HDLs were eluted with ApoA-1 (Fig 2B and S4B Fig in S1 File). Fractions of LAC1 and LAC2 eluted with major antiproteases, Alpha-2-macroglobulin and inter-alpha-trypsin inhibitor, respectively, similar to those in rats (Fig 2C and S4C Fig in S1 File) [9]. Antiproteases, and thus LACs, are synthesised in the liver

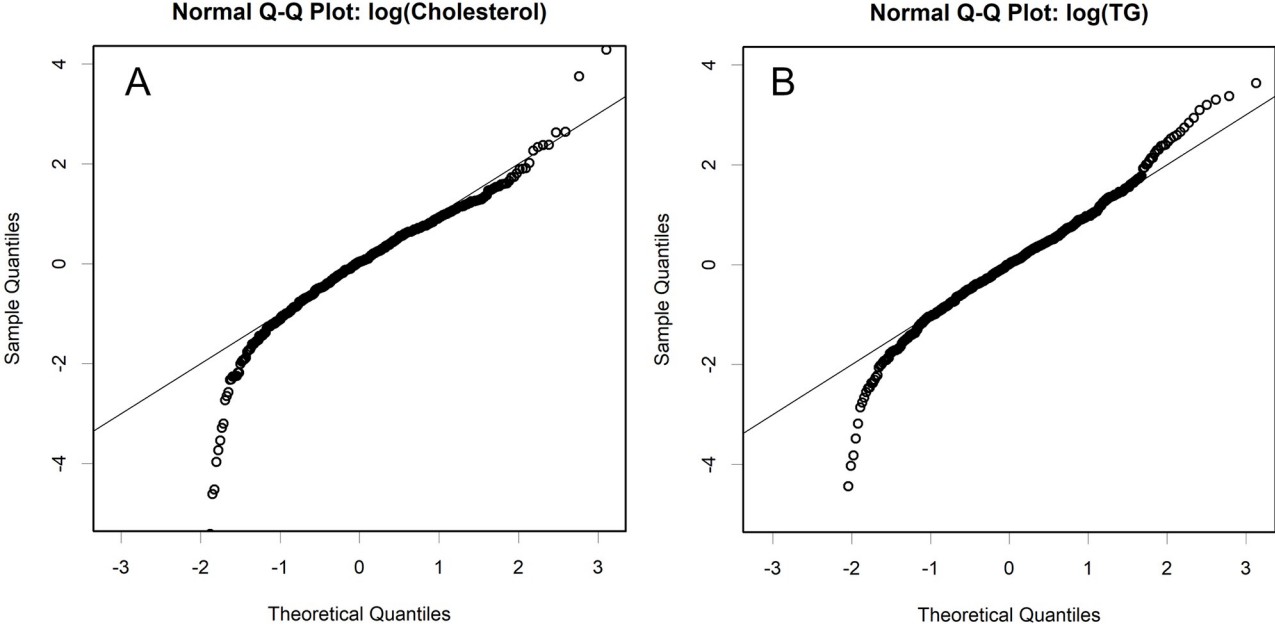

**Fig 4. Normal QQ plot of the logarithms of normalized data.** Cholesterol (**A**.) and TG (**B**). Quantiles are compared between the theoretical normal distribution and the z-normalized log-data. Here a lognormal distribution will form a straight relationship.

[16,17]. There were two additional classes of VLDL not observed in the rats: a class with Apolipoprotein (a) and a class composed almost entirely of TG.

There were some differences observed between rat and human lipoproteins. There were three classes of VLDL particles in human lipoproteins, which contained ApoB100 (Figs 1 and 2A, and S4A in S1 File), compared to rat lipoprotein which only has one class of VLDL particles. The largest class is denoted as VLDL here; this contains more TG than cholesterol. Slightly smaller than this, very cholesterol-rich class included another lipoprotein denoted as Lp(a) protein [18,19] (Fig 4A). The size of the particles seems to be larger than that previously reported [18]; ultracentrifugation may have removed a portion of the particles. Another smaller class was noted to have a size slightly larger than that of LDL1. Estimating the cholesterol content of this class was difficult as its size is close to that of LDL1. However, as the fitting results suggested low cholesterol (S2 Table in S1 File), this is denoted as the TG-rich class (TR) in this study. These classes showed no relationship (S5A and S5B Fig in S1 File). Rather, the cholesterol carrier, Lp(a), showed a weak negative correlation with the levels of another carrier, LDL2 (S5C Fig in S1 File). Incidentally, there is a positive correlation between LP(a) and LDL1 cholesterol, suggesting that LP(a) is one of the precursors of LDL1 (S5D Fig in S1 File).

There were two sizes of larger particles with the largest particles with B48 denoted as CM1. The 50–60 nm particles had B48 (Figs 1 and 2A, S3A Fig in S1 File), and the level showed a higher correlation with CM1 than VLDL (S5E and S5F in S1 File); hence, the class is denoted as CM2. These particles would be intestine-derived [7,8].

The amount of each class varied significantly among the volunteer samples (Fig 3). They were lognormally distributed, as was found from the logarithms of the data quantiles, which showed a linear relationship with the theoretical values of the normal distribution. Linear correlations were confirmed in all classes (S6 Fig in S1 File); The slope and intercept of the regression line represent σ and μ of the normal distribution, respectively; these are the distributions used to fit the curves of each sample. The parameters were estimated by robust methods,

specifically MAD and trimmed means; the appropriateness of the estimation can be checked to determine whether the lines fit the plots. By accumulating z-normalized data using these parameters, $z_i = (\log(x_i) - \mu) / \sigma$ for any raw data $x_i$, the distribution can be confirmed in a more exact quantile-quantile (QQ) plot (Fig 4), where the slope is 1 and the intercept is zero. The lower part of the QQ plot bends downward, which is considered to be an artefact during curve fitting because low values are easy to ignore.

Knowing that the data is lognormally distributed, the standard value and the 95% range of the classes could be estimated with accuracy, even though they were largely dispersed (S2 Table in S1 File). Note that the interval ranges inevitably become asymmetric to the standards: the higher is always wider.

The ratio of TG to cholesterol in each class also fluctuated greatly (Fig 3A). Naturally, these ratios are large in the class synthesised to include TG, such as CM and VLDL. HDL1 and 2 are considered precursors to mature form of HDL (mHDL) (Fig 2B and S4B Fig in S1 File). They mature by receiving cholesterol.

LDL1 was the most predominant cholesterol carrier, followed by LAC2 (Fig 3B). Some cholesterol carriers appeared within certain narrow ranges (LDL1, LACs, and HDLs). These classes may maintain cholesterol homeostasis. Conversely, variations in CM and VLDL levels were particularly high (Fig 3B). The former may depend on diet, and the latter may depend on the body's requirements for TG as a storable energy source. Among the TG-rich classes, TR appeared to be fairly stable.

HDLs were minor cholesterol carriers, contradicting the estimations of the previous study, which did not check whether each of the fractions contained ApoA-1 protein [20]. In the study by Gordon et al., they assigned the whole of a major peak of cholesterol as HDL, which seemed to be a reasonable decision, since HDL was believed to be the major cholesterol carrier from ultracentrifugation results [5,7,8]. However, in reality, the peak size is too large to be assigned to HDL alone; the structure of HDL is surrounded by ApoA-1 and therefore has a limitation in size [9,21]. In fact, ApoA-1 was only confirmed at the end of the peak (Fig 2B and S4A Fig in S1 File). Rather, the peak was mainly composed of LDL2 and LAC2 (Fig 1); these classes may have behaved as high-density particles during ultracentrifugation. However, their origins and functions are quite different from those of HDL.

The amount of protein was estimated using the bicinchoninic acid assay, and the ratios of the lipids present (TG and cholesterol) were observed (Fig 4D). In both VLDL and CM classes, the TG and cholesterol levels varied considerably when compared to the protein levels; however, they always contained fewer lipids than protein, contradicting the results of ultracentrifugation [7–9]. Serum contains a large amount of albumin, and human serum samples have a larger number of immune-related proteins than rodent samples (S7 Fig in S1 File). However, this fact did not affect the estimation of the CM and VLDL content. Single proteins that did not vary in size were eluted using HPLC with a sharp normal distribution. Albumin was detected in fractions 27 and 28, close to HDL (Fig 1), and did not cross over into other fractions. Some of the immunoglobulins have a comparable size to some lipoproteins; however, as CM and VLDL are very large particles, the corresponding fractions would be free from albumin and immunoglobulins.

The amounts of HDL and LDL measured by conventional enzymatic methods were much higher than those of any of the classes. This is not surprising, as those account for the majority of total cholesterol (S8 Fig in S1 File). If conventional methods extract cholesterol from certain classes with high efficiency, the observed data would reflect combinations of some classes. Such combinations were estimated as combinations with the highest correlation and similarity (Fig 5). The conventional measure of LDL would be VLDL derivatives without VLDL, and HDL would be HDLs and CM derivatives without CM1. Of course, we cannot deduce any of

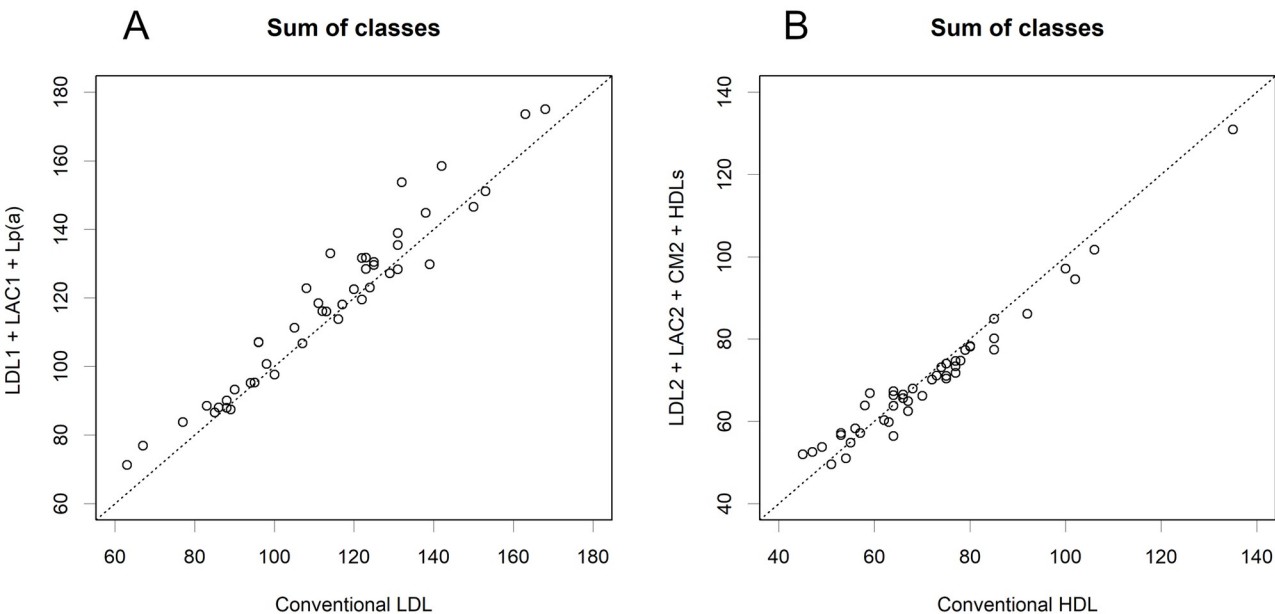

**Fig 5. Estimation of the sets of classes that are measured by a conventional method.** If the conventional methods extract most of the cholesterol from particular classes, these would present the sum of the quantities of the classes. The combinations presented here were the ones with the highest correlation and the closest amounts. **A**. Conventional LDL. Pearson's $r = 0.9680$; $p < 2.2e\text{-}16$. **B**. Conventional HDL. Pearson's $r = 0.9806$; $p < 2.2e\text{-}16$.

the true classes from the conventional measurements. In particular, the estimated HDL differed significantly from the actual amount of HDLs. Moreover, sums of lognormally distributed quantities are not informative at all, as an anomalous class may make up the majority of the sum; in a statistical sense, they do not even show average levels (S9 Fig in S1 File). Unfortunately, it is also true that total TG or cholesterol does not provide useful information. Rather, an exact measurement of each class is desirable.

## Discussion

Many features in human samples were similar to rat results [9]. Each elution pattern was a mixture of normal distributions (Figs 1 and 2, and S2-S4 Figs in S1 File). This shows that each class of lipoprotein is stable in serum, which is inconsistent with the scenario in which CM and VLDL lose TG by gradual degradation [5,7,8], which in turn produces more skewed distributions [9]. Rather, a single degradation of TG should have converted them into the next class at once. Humans also express the LDL-antiprotease complexes (LAC) [9]. Furthermore, HDL was a rather minor cholesterol carrier, and the major component of lipoproteins, as far as can be ascertained, was protein, not lipid. For a detailed discussion of these details, please refer to the previous study in rats [9].

However, humans differ from rats by having three types of VLDL-like particles. These differ greatly in the cholesterol/TG ratio (Fig 3); the most-cholesterol-rich was Lp(a) and the most-TG-rich was TR. Since they were each present in unrelated amounts, their biosynthesis is probably differentially regulated (S5 Fig in S1 File). Humans may be producing the appropriate class according to the body's requirements.

The particle size and ApoB protein distribution suggest that they can be classified into three lineages:

1. CM → LDL2 → LAC2

2.  VLDL, Lp(a) → LDL1 → LAC1

3.  HDL2 → HDL1 → mHDL.

These protein distributions were the same as those of the rats [9]. In this study, we focused on determining the attributes; hence, we did not analyse other apoproteins such as ApoC or ApoE. Certainly, we may only identify a fraction of apoproteins [9]. Additionally, it should also be noted that the number of classes presented here was the smallest to perform curve fitting (S2 Fig in S1 File). Systems with better separation may find more classes. The positions and scales of the classes are listed in S1 Table in S1 File. Classes that were initially secreted from the liver or intestine showed a larger scale. The particles may have a larger tolerance for the ratio and amount of lipids, which may have expanded this parameter. However, the elution position of the class did not change significantly between the samples. Therefore, the estimated peak diameters were within a certain range, indicating the physical stability of the particles (S1 Table in S1 File).

It is not surprising that each class was lognormally distributed. For example, the transcriptome has the same distribution properties [22]. Levels of lipoproteins, as well as mRNA, are regulated by a balance between synthesis and degradation, both of which are controlled by specific factors in a multiplicative manner; this mechanism determines the distribution (S10 Fig in S1 File). Therefore, lipoprotein levels can easily change due to multiple small causes; differences in multiple lifestyle habits may worsen the medical condition. In contrast, simultaneous clinical efforts may synergistically improve lipoprotein levels. This would explain why a different set of feeds drastically altered lipoprotein levels in a rat study [23].

In contrast to rats, human serum contained higher amounts of immunoglobulins. It should be noted that rats grown in pathogen-free environments did not show detectable levels of these proteins in SDS-PAGE [9]. Some of them were as large as certain classes of lipoproteins (S7 Fig in S1 File), which interfered with the estimation of the protein abundance in these classes, which were able to be estimated in rat samples [9].

When separating lipoproteins by HPLC, it can be difficult to perform curve fitting due to the close size of some classes, depending of course on the column characteristics. In our case, LDL2 and LAC2 were both major and similar in size, and their peaks overlapped to form a composite peak. The minor mHDL, which is similar in size to these, likely leads to a larger measurement error. Nevertheless, they were ultimately measured within a certain range here (Fig 3). However, when the quantitative balance among these classes is disrupted by disease, the estimation of some classes by curve fitting may become difficult. Actually, in Fig 4, the lower side of the lognormal distribution always decreases, which is likely a result of not being able to fit the curves for lesser classes and ignoring them.

The data analysis method used in this study is different from the previous methods; we parametrically analysed and divided the peaks and demanded biochemical confirmation for each assignment. This made the results quite different from previous results, which were based on ultracentrifugation analysis. What stands out most clearly is the small amount of HDL. In HPLC, the cholesterol curve has two main peaks; in the past, they were considered to be LDL and HDL. However, the latter peak was a combination of several LDL and HDL peaks. If we assume that all these peaks are HDL by neglecting the evidence displayed in Fig 2B, the results would be similar to the ultracentrifugation results, which is probably why the error occurred. The same phenomenon occurs in NMR, which assigns too many peaks to HDL [24]. In contrast to HPLC, which separates peaks by size, peak assignment in NMR was performed by the fitness of complex spectrum data obtained at various temperatures and pressures to the ultracentrifugation results. Therefore, peak assignment has been influenced by incorrect results. Fundamentally, this is because we do not know the specific spectrum of cholesterol in the

presence of various classes of lipoproteins. NMR is still a promising approach, but the peak assignment needs to be revised using accurate information.

Lipoproteins were probably degraded during the ultracentrifugation process, which has long been the de facto standard method of analysis but has a greater impact on the specimen than HPLC. Ultracentrifugation has many variations; however, it is basically a sequential combination of three centrifugations [1–5]. In typical density gradient centrifugation, a density gradient is formed in advance using non-ionic dense materials, such as sucrose or colloidal silica solutions. They have the density to float any lipoprotein and can reduce the required time; however, this method has not been used for lipoproteins. In isopycnic centrifugation, a salt concentration gradient is formed spontaneously by centrifugal force. Inevitably, the sample was exposed to a fairly high salt concentration (2–7.7N NaBr) for a long time. This concentration is sufficient to salt-out many proteins [8]. Eventually, the fractions that floated in the solution were sequentially collected to isolate progressively denser classes from CM to HDL. The density gradient takes approximately a day to form when the samples are exposed to endogenous proteolytic enzymes. The centrifugal force was 500,000 g. This is sufficient gravity to sediment the proteins against Brownian agitation, and it is common practice to estimate molecular weight from the sedimentation coefficient measured in this way [8].

On the other hand, HPLC is a method of separation by size; therefore, if particles of the same size are present, they become contaminants. However, CM and VLDL were > 50 nm in diameter (Fig 1). These are very large, which is evident from the fact that ribosomes are less than 30 nm in diameter. The serum does not contain cellular components such as platelets, and the presence of such large protein complexes other than lipoproteins is rare. Another protein, alpha-2-Macroglobulin, is a hydrophilic glycoprotein, and if we assume a sphere with an average density, this 720 kDa protein should be approximately 12 nm in diameter. The size of LAC is 18 nm (Fig 1); few factors other than LDL binding could cause this size increase, and the overlap of the peak with cholesterol (Fig 2B) is probably not coincidental.

HPLC and ultracentrifugation differ greatly in the composition and density of CM and VLDL. The amount of protein was nearly 90% in HPLC (Fig 3D), but only trace amounts in ultracentrifugation [8]. As the average protein density is approximately 1.43, the density of the particles, of which 90% is protein, should be approximately the same. However, the densities of CM and VLDL separated by ultracentrifugation are approximately 1 [8]. This difference is understandable if the molecules are decomposed during ultracentrifugation. The centrifugal force from gravity on a 150 nm diameter protein is 12 pN, and the buoyancy force is approximately 8 pN; therefore, this particle will move downward with a force of 4 pN. If the particle remains in place, this force pulls the protein. Aquaporins and rhodopsins, which are typical hydrophobic proteins with transmembrane domains, can be pulled out of the membrane when subjected to forces of tens to 100 pN [25–27]. In addition to hydrophobic proteins such as ApoB, more loosely bound proteins could be released with even less force. Although many of the forces that have been measured with atomic force microscopy so far are limited to fairly strong bonds, such as avidin to biotin or antibody to its target, little is known about the reality of weak protein-protein interactions [28]. Even for a single fluctuation, once the bonds are dissociated, and the objects are captured by gravity, they are pulled apart. This is because the concentration of these materials during centrifugal separation is so low that an equilibrium state cannot be assumed between dissociation and binding. In addition, because salting-out proteins are insoluble in water, they will sediment with even more force. In contrast, HPLC can complete the separation in tens of minutes with a mild buffer, and it does not apply much force that could rupture the molecules.

Thus, the perception of lipoproteins should be updated. They are classes of protein-rich particles, each of which has specific functions. Only limited apoproteins were studied here, but

the attributes of other apoproteins are important for their functions. Hence, the classes need to be studied extensively, which will provide a deeper understanding of the pathophysiology. The levels of the classes varied among volunteers and were lognormally distributed. Some classes are narrowly controlled and are good candidates for indicators of diseases. Conventional enzymatic methods measure mixtures of multiple classes. We had much less HDL than previously believed. Because the sums of lognormally distributed numbers are not informative (S9 Fig in S1 File), the data do not provide a proper diagnostic criterion. This could be the reason why levels of conventional LDL did not indicate the prognosis of the patient [29]. However, HPLC is not suitable for large numbers of samples. Therefore, simpler methods for measuring specific classes are required. In particular, the targets of enzymatic methods should be completely refined; currently, we are not able to measure the classes for which each is intended (Fig 5). So far, these methods have been used not only in health screening methods but also in many studies. The current knowledge on lipoproteins, such as the levels of *healthy* HDL or LDL derived from these studies should, unfortunately, be re-evaluated using HPLC since no alternative exists. According to the lognormal distribution characteristics, independent clinical treatments, such as antilipidemic drugs, nutritional therapy, and ergotherapy, may synergistically change the levels of specific classes.

## Supporting information

**S1 File.**
(ZIP)

## Acknowledgments

We would like to thank Editage (www.editage.com) for English language editing.

## Author Contributions

**Conceptualization:** Tomokazu Konishi.

**Data curation:** Risako Fujiwara, Yoshihisa Abe, Shinpei Kawata.

**Formal analysis:** Tomokazu Konishi.

**Investigation:** Tomokazu Konishi, Risako Fujiwara, Tadaaki Saito, Nozomi Satou, Yurie Hayashi.

**Methodology:** Tomokazu Konishi.

**Project administration:** Tomokazu Konishi, Tatsuya Ishikawa.

**Software:** Tomokazu Konishi.

**Supervision:** Tomokazu Konishi, Naoko Crofts, Ikuko Iwasaki.

**Visualization:** Tomokazu Konishi.

**Writing – original draft:** Tomokazu Konishi.

**Writing – review & editing:** Tomokazu Konishi.

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
