## [Decision Letter · Decision Letter 0]

22 Apr 2022

PONE-D-22-03128Human lipoproteins comprise at least 12 different classes that are lognormally distributed.PLOS ONE

Dear Dr.Tomokazu Konishi

Thank you for submitting your manuscript to PLOS ONE. After careful consideration, we feel that it has merit but does not fully meet PLOS ONE’s publication criteria as it currently stands. Therefore, we invite you to submit a revised version of the manuscript that addresses the points raised during the review process. Please ensure your supplemental material can be easily identified.

We look forward to receiving your revised manuscript.

Kind regards,

Jérôme Robert, PhD

Academic Editor

PLOS ONE

“TK received a grant from Akita Prefectural University.”

We note that you have provided additional information within the Funding Section that is not currently declared in your Funding Statement. Please note that funding information should not appear in the Funding section or other areas of your manuscript. We will only publish funding information present in the Funding Statement section of the online submission form.

Reviewers' comments:

Reviewer's Responses to Questions

**Comments to the Author**

1. Is the manuscript technically sound, and do the data support the conclusions?

Reviewer #1: No

Reviewer #2: Yes

2. Has the statistical analysis been performed appropriately and rigorously? 

Reviewer #1: No

Reviewer #2: Yes

3. Have the authors made all data underlying the findings in their manuscript fully available?

Reviewer #1: Yes

Reviewer #2: Yes

4. Is the manuscript presented in an intelligible fashion and written in standard English?

Reviewer #1: Yes

Reviewer #2: Yes

5. Review Comments to the Author

Reviewer #1: In this work, Tomokazu and colleagues provide an alternative classification of lipoprotein subclasses in human plasma after HPLC-based separation. The conclusions that are drawn by the authors, despite interesting, clash against multiple well established aspects of lipoprotein metabolism and thus should be supported by a larger and more detailed corpus of data, which is not present in the current work. In particular the manuscript lacks any mechanistical data supporting the key claims about e.g. HDL cholesterol content as well as TG-rich lipoprotein catabolism. My comments are as follows:

Methods:

Line 210: “No postmeal time was specified for blood collection.” Is repeated twice

Line 211: “The age of the volunteers and the sex of the subjects were 50/50, as shown in Figure S1”. This statement is in my opinion unclear. On the one hand, there is not information regarding sex in Figure S1. What is the female to male ratio in the study? How many females/how many males? Additionally, what is meant here with “50/50”? Was the mean age in the study 50 years?

Line 214: here “HDL” should be used instead than “HLD”. Also please indicate what were the “conventional methods” used to determine them. E.g. what commercial kits were used

Line 216: What are the parameters used in the HPLC analysis? What column? What flow rate? What medium? Where different conditions used for analytical VS preparative HPLC? What methods were used to monitor TG and cholesterol?

Line 217: what do the authors mean here by hyperlipidaemic? The plasma lipid parameters of these patients as well as the type of hyperlipidaemia should be specified in the manuscript

Line 225: “class., with The amount”. Take dot away. Do not capitalize T in “The”

Line 236-237: please either specify the MALDI-TOF-MS methods or provide a reference to a previous work where they were used.

Line 243: “which has better sensitivity than the biuret reaction commonly used for serum samples”. This assertion is not correct, as the BCA method is based on the biuret reaction as well. Do the authors intend that the BCA method is more sensitive compared to the Bradford method? If yes, they should change this sentence accordingly

Figures:

Fig 1. As far as I understand, in the figure two curves are shown that display the raw data for TG as well as for cholesterol. Both are shown in gray, which in my opinion is quite confusing, as curve-fitted data for TGs and choelesterol are then shown in red and blue. Curves for raw data should also be colored (e.g. in a different shade of red and blue), to indicate whether they refer to TG or cholesterol.

Additionally, the SDS-PAGE gel in the left panel should be shown for clarity in a separate panel within the same figure. The SDS-PAGE panel should also indicate what are the numbers on the axes referring to (Molecular weight and elution time?)

Last, in the SDS-PAGE the ApoB100 peak seems to reach its maximum between 10 to 19min, while no ApoB is visible after the 21 minutes fraction. Nevertheless, the authors depict a high “LDL1” peak with mean elution time of about 23 minutes. Could the authors elaborate on this? It is especially difficult to reconcile the band intensities in this silver-stained gel with the ApoB quantification data should in Fig S3, which show instead a clear ApoB100 peak around 23 minutes. A similar discrepancy between Figure 1 and S3 seems to affect the elution curve of ApoB48 as well.

Figure 2B: (see also comment to Fig S6). Based on this graph, the HDL fractions combined seem to contain much less cholesterol compared to the LDL fractions than expected for “normal” plasma LDL-C and HDL-C values. The authors should elaborate on this point.

Fig S2: the abbreviation “TR” should be explained again in the corresponding line of the Results section (line 61) as well as in this figure’s legend. A definition is given instead only in line 80 of the results. Additionally, measurement units should be indicated on the y axes in both panels. The expression “It is unique in that it is smaller and very low in cholesterol. However, fitting is difficult without this.” Is quite unclear without having read line 80.

Fig S3: please see comment given for Fig 1. The authors should indicate what the two panels shown in this figure indicate. Are they curves obtained from two different samples? This should be mentioned in the legend. Additionally, what to the small gray numbers in the middle of each panel indicate?

Fig S6 The authors show a “LDL2” fraction, which nevertheless seems to appear clearly after the bulk of ApoB100 has been eluted. The HDL-cholesterol peak seems also to be much smaller than expected for an individual with an average “healthy” lipoprotein profile. Could the authors elaborate on this? How can the authors be sure that the “LDL2” peak does not in fact depict the main HDL-cholesterol peak? Additionally, what does the number “401” in the left panel indicate? In this figure as well, the SDS-PAGE gel should be shown as a separate panel, indicating what is shown in each axis.

Fig S8: the authors should provide correlation coefficients and p-values for linear regression for each panel.

Results:

Lines 62-63: “The attributes of the classes were determined based on the elution pattern of the major protein components (Fig. 1, S3-S6). The elution patterns of these proteins coincided with the distribution of the corresponding classes”. If the classes were assigned based on the elution pattern of the major protein components, it would then be logical that they would elute at the same time. The authors should thus delete the second statement. Also, please see my comments regarding figures 1, S3 and S6 on apparent incongruences regarding the elution times of different proteins and lipoprotein subclasses

Line 63: “As the positions and scales coincided between TG and cholesterol”. The meaning of this sentence is unclear to me. Do the authors intend to say that cholesterol and TG followed a similar elution pattern within each subclass, thus suggesting a stable cholesterol-to-TG ration within that class?

Line 68: “and the LDL2 fraction is devoid of B48”. Please see my comments on Figures 1, S3 and S6. Besides being devoid of ApoB48, this fractions seems not to contain ApoB100 either based on the data in Fig S3, which would clash against their classification by the authors as LDL

Line 71: “they may supply cholesterol to the thrombus and protect it from lysis by plasmin”. This statement belongs in my opinion to the discussion section. Additionally, the authors should indicate on what this statement is based (other data? literature?)

Lines 80-83: the authors should provide a statistical evaluation of correlation of the parameters analyzed in Fig S8, including correlation coefficients and p-values.

Line 114: “HDLs were minor cholesterol carriers, contradicting the estimations of the previous study”. I find this statement problematic, and the reference to Gordon et al., (doi: 10.1021/pr100520x) who show how the ApoA-I-containing HDL fractions also carry large amounts of cholesterol in normolipidemic patients as contradicting the findings of the current study. The fact that Gordon and colleagues did not measure ApoA-I levels in each HDL fraction doesn’t cancel the fact that they nevertheless identified it as the most abundant HDL-associated protein in their MS experiments.

Line 118: “the structure of HDL is surrounded by ApoA-1 and therefore has a limitation in size”. The authors refer to Wu et al., doi: 10.1074/jbc.M110.209130. The authors should elaborate more on why the structural findings on this paper set a limit on HDL size based on the conformation of ApoAI.

Lines 123-126: these data seem in contrast, among others, with those of Gordon et al (see comment above). The authors should explain how their findings compare to other HPLC-based studies in this regard.

Discussion:

Lines 149-152: these conclusions contradict data about lipoprotein metabolism gathered through multiple decades and should thus be supported by a much larger amount of evidence as well as a more detailed interpretation of the data. Among others, the selective modification/knockdown of key players in each lipoprotein metabolism pathway in mice as well as genetic studies in human mutation carriers have allowed to establish the current model of lipoprotein metabolism, which, despite not perfect or complete, cannot be discarded as quickly as done in this section of the current manuscript. Thus, the authors should attenuate their claims and provide them in the context of what is known e.g. in relation of the catabolism of TG-rich lipoproteins in humans.

Lines 148-186: please see my previous comments about how cholesterol peaks were assigned to each subclass.

Reviewer #2: The manuscript describes the results of a rapid HPLC method to separate human lipoproteins. The authors identify 12 lipoprotein classes log-normally distributed. The study technically sounds and the main conclusions are supported by data. I have no major concerns but some points should be addressed.

The manuscript is hard to follow. Many data are included in the supplementary figures, this reviewer tried to download these suppl data and it was impossible to identify the different figures cited in the text.

The levels of VLDL, Lp(a), and TR (triglyceride-rich lipoproteins) showed no relationship, suggesting that they were produced independently (Fig. S8AB). I cannot download this figure, but it is difficult to believe that the levels of VLDL and TR were produced independently. How? VLDL is one of the main determinants of TR.

Define mHDL (mature HDL?)

Since there is significant overlap in several lipoprotein classes, TG and cholesterol distribution for these lipoproteins is a critical point. This point should be clearly addressed.

6. PLOS authors have the option to publish the peer review history of their article (what does this mean?). If published, this will include your full peer review and any attached files.

Reviewer #1: No

Reviewer #2: No

---

## [Author Response · Author response to Decision Letter 0]

2 Jun 2022

please see the file attached, "Response to Reviewers.docx".

---

## [Decision Letter · Decision Letter 1]

11 Aug 2022

PONE-D-22-03128R1Human lipoproteins comprise at least 12 different classes that are lognormally distributed.PLOS ONE

Dear Dr. Konishi,

Thank you for submitting your manuscript to PLOS ONE. After careful consideration, we feel that it has merit but does not fully meet PLOS ONE’s publication criteria as it currently stands. Therefore, we invite you to submit a revised version of the manuscript that addresses the points raised during the review process.

We look forward to receiving your revised manuscript.

Kind regards,

Jérôme Robert, PhD

Academic Editor

PLOS ONE

Journal Requirements:

Additional Editor Comments:

Dear Authors,

After carefully reading your manuscript, the academic editor and two reviewers recommend publication of your work in PLOS One.

According to the new reviewer we ask you to address the following two points:

1. In the last paragraph of the introduction, the authors mention that ultracentrifugation causes loss of membrane proteins. It is an important detail for the present story. The authors should check the reference they have added here and provide more references, if available, supporting this claim.

2. In the discussion section, the authors appear to presume a wider role of the different LPP. Although plausible, this is not part of the present work. It would be good that the authors lower the claims and describe more on the results from the present work.

Best regards

Reviewers' comments:

Reviewer's Responses to Questions

**Comments to the Author**

1. If the authors have adequately addressed your comments raised in a previous round of review and you feel that this manuscript is now acceptable for publication, you may indicate that here to bypass the “Comments to the Author” section, enter your conflict of interest statement in the “Confidential to Editor” section, and submit your "Accept" recommendation.

Reviewer #2: All comments have been addressed

Reviewer #3: All comments have been addressed

2. Is the manuscript technically sound, and do the data support the conclusions?

Reviewer #2: Yes

Reviewer #3: Yes

3. Has the statistical analysis been performed appropriately and rigorously? 

Reviewer #2: Yes

Reviewer #3: Yes

4. Have the authors made all data underlying the findings in their manuscript fully available?

Reviewer #2: Yes

Reviewer #3: Yes

5. Is the manuscript presented in an intelligible fashion and written in standard English?

Reviewer #2: Yes

Reviewer #3: Yes

6. Review Comments to the Author

Reviewer #2: The authors addressed the concerns of this Rw. I have no further comments.

Maybe the authos could add subsections to facilitate the reading of the paper.

Reviewer #3: In the present study, Konishi et al, aim to identify a wider range of LP particles exists in human plasma. The premise of the study is that the currently established and widely used methods of LPP purification or separation alter the protein composition or cause degradation of particle constituents that would eventually affect adversely the class separation or identification. They previously identified 10 different classes of serum LPP in rats and now show the existence of 12 such classes in humans. They use lipid quantification, Western blotting and protein sequencing for identification of each class post HPLC separation. The results, in general, are interesting and the MS that this reviewer received is an improved version over the first submission from the authors. These observations are intriguing given the importance of LPP in metabolic disease.

I have, however, some minor points for the authors.

1. In the last paragraph of the introduction, the authors mention that ultracentrifugation causes loss of membrane proteins. It is an important detail for the present story. The authors should check the reference they have added here and provide more references, if available, supporting this claim.

2. In the discussion section, the authors appear to presume a wider role of the different LPP. Although plausible, this is not part of the present work. It would be good that the authors lower the claims and describe more on the results from the present work.

Barring these minor points, I recommend publication of this work in PLOS ONE.

7. PLOS authors have the option to publish the peer review history of their article (what does this mean?). If published, this will include your full peer review and any attached files.

Reviewer #2: No

Reviewer #3: No

---

## [Editor Report · Decision Letter 2]

12 Sep 2022

Human lipoproteins comprise at least 12 different classes that are lognormally distributed.

PONE-D-22-03128R2

Dear Dr. Konishi,

We’re pleased to inform you that your manuscript has been judged scientifically suitable for publication and will be formally accepted for publication once it meets all outstanding technical requirements.

Kind regards,

Jérôme Robert, PhD

Academic Editor

PLOS ONE

---

## [Editor Report · Acceptance letter]

6 Oct 2022

PONE-D-22-03128R2 

Human lipoproteins comprise at least 12 different classes that are lognormally distributed. 

Dear Dr. Konishi:

I'm pleased to inform you that your manuscript has been deemed suitable for publication in PLOS ONE. Congratulations! Your manuscript is now with our production department. 

Kind regards, 

on behalf of

Dr. Jérôme Robert 

Academic Editor

PLOS ONE